

# Development and feasibility study of segment blade test methodology

Kwangtae Ha[1], Moritz Bätge[1], David Melcher[1] and Steffen Czichon[1]

[1]Fraunhofer Institute for Wind Energy Systems, Department Rotor Blades, Bremerhaven, 27572, Germany

*Correspondence to*: Kwangtae Ha (kwangtae.ha@iwes.fraunhofer.de)

**Abstract.** This paper outlines a novel segment test methodology for wind turbine rotor blades. The segment test mainly aims at improving the efficiency of the fatigue test as a future test method at Fraunhofer IWES. While resulting in reduced testing times, target fatigue bending moments over the intended test area have to be matched within an acceptable range of overload. The numerical simulation reveals that the proposed segment testing has a significant time saving up to 43% and 53% for 60m and 90m blades. When compared to the experimental data of 60m full-length blade fatigue test, the proposed methodology also shows better test quality over the intended area and better certifiable regions over the outboard area.

## 1 Introduction

Wind turbine rotor blades are designed to sustain the designed 20 ~ 25 years lifetime without structural failure. Reliability of the blade is essential for keeping maintenance and operations costs low and maintaining the designed power performance. Mechanical full-scale tests are the main process available for validating the structural integrity of wind turbine rotor blades (Spera, 2009). This is also required as part of the blade certification process according to IEC 61400-23 (IEC, 2012). A typical full-scale blade test consists of following steps:

- Determining blade properties such as mass, center of gravity, natural frequencies, etc.
- Static load test
- Fatigue load test
- Static load test after fatigue test

Among all structural tests, the fatigue load test is the most time consuming process, and therefore this step accounts for most of the test cost involved. It is executed either in the flap or lead-lag direction independently, or simultaneously in both directions. Due the large mass and low stiffness of the blades, test operating frequencies are very low, especially for flap fatigue tests. Therefore, fatigue tests take up to several months to accomplish the required cycle count of 1 to 5 million cycles, depending on the blade characteristics (Wingerde et al., 2015; Post, 2016). As blades are getting longer and bigger, both test time and costs increase significantly - especially for future offshore blade measuring more than 100m in length. Due to the high associated investment costs, only few test rigs are expected to be able to sustain the large bending fatigue bending moments for testing 100m+ blades. Hence, a reduction in test time is required to prevent the availability of test rigs to become a bottle neck for turbine development.






In this paper, a novel segment test methodology for wind turbine rotor blades is proposed and its benefits are investigated with a numerical simulation. The proposed test methodology decouples a blade root segment and a tip segment to improve test quality and minimize test time by performing both segment tests independently in parallel. Two representative blade models, a 60m onshore blade and a 90m offshore blade were used for numerical simulation to compare the fatigue test time and the

test quality of segment blades with full-scale blades in ANSYS APDL and ANSYS Workbench 18.1 (ANSYS, 2018).

## 2 Segment blade test methodology

Since the tip portion of the blade is generally not very critical in terms of structural damage, the blade tip is sometimes removed during fatigue testing in order to reduce energy consumption and test time. This can also be done if space in the testing facility is limited. (Spera, 2009; Al-Khudairi et al., 2017; White, 2004). The segment blade test takes this approach a step further and

divides the Device under Testing (DUT) into a root and a tip segment as shown in Fig. 1, which can then be tested in parallel.

### 2.1 Differences between full-length and segment testing

The test procedure remains essentially unchanged compared to the full length blade test. However, there are several important differences in the test preparation and execution that have to be considered.

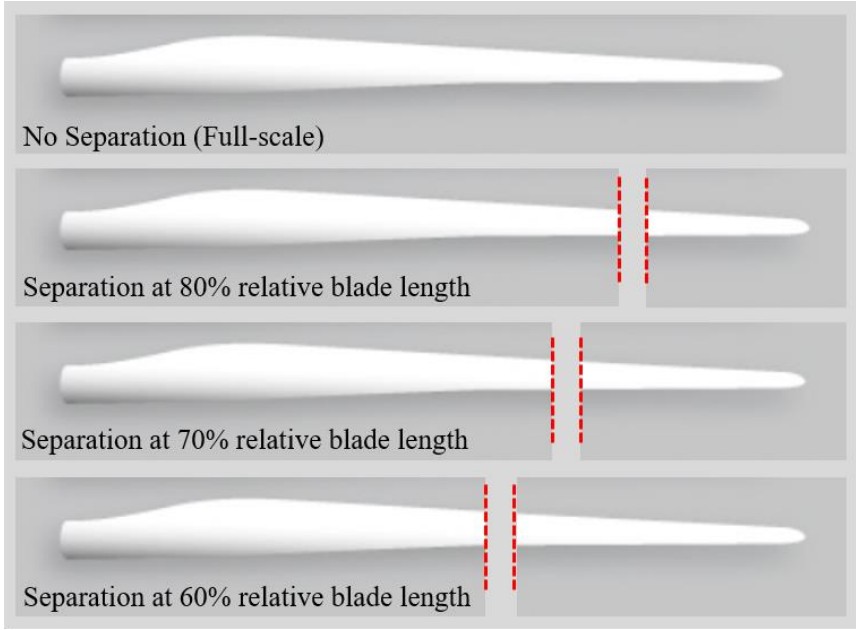


**Figure 1.** Schematic representation of a blade with different separation points





It is assumed that design-critical areas are not located near the separation points. If the areas of the separation point are to be verified in the test, a significantly greater manufacturing outlay arises, as overlapping segments have to be produced. In this
work, the assumption applies that the separation point does not lie in one of the critical areas (Brondsted et. al, 2013). Special care has to be taken when cutting the blade to produce reliable segment blades without any defect. However, this aspect is not considered in this study (Skelton, 2017).

The expense of equipping the rotor blade with test-specific measurement sensors does not change significantly as a result of
the segmentation. However, since the blade tip test time is much shorter than the blade root test due to the higher test frequencies, the preparation work for the tip segment can be postponed. In other words, after the blade root test is prepared and started, preparation work with the blade tip segment can be executed. In this way, additional time can be saved in the overall process of the test.

In order to improve transportability, some modern onshore blades are already produced in a modular design, where two halves are produced that are joined on the wind farm site. For these blades, the tip already has a connection to mount the tip segment to the test stand. If the blade is only separated for test purposes, the tip segment, unlike the root segment, has no prefabricated connection for mounting to the test stand. In this case, the mounting test stand must be fabricated for the tip segment. There are several possible solutions. One solution is that the segment can clamped by two fixed load frames arranged one behind the
other. However, this leads to a reduction of the test area. Fraunhofer IWES has developed and validated an alternative fixture concept. The structure to be tested is glued firmly into a wooden frame, whereby the immediate connection point is reinforced by steel sheets (Rosemeier et. al, 2018). Fig. 2 (a) shows this connection technology used in the application of a trailing edge subcomponent test, and Fig. 2 (b) shows the connection design of the tip segment to the test bench as part of the entire test setup without additional masses. Depending on the length of the tip segment, the bonding depth to the load frame may vary to
sustain the maximum strength at the clamping area.

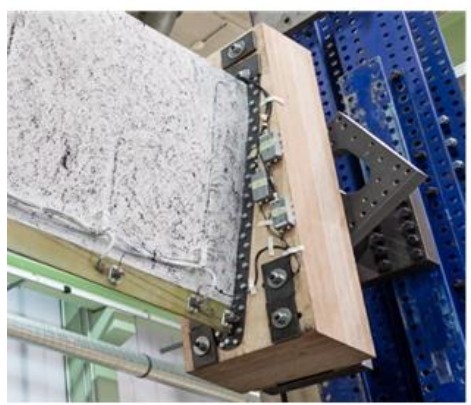
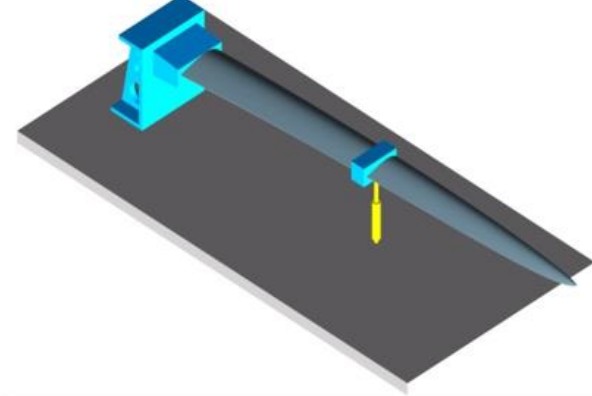

(a) Load frame for blade subcomponent test        (b) Connection design for blade tip segment

**Figure 2.** Solutions for mounting blade tip segment to the test stand





During the conventional fatigue test, the rotor blade is usually more stressed in the area near the root than in the area of the
blade tip compared to the target loads. As an example, Fig. 3 shows an experimental overload distribution of a 60m blade in
the lead-lag fatigue test. As a result, the inboard areas of the rotor blade have already experienced the calculated loads from
20 years of life, but the test must continue to sufficiently load the under-stressed areas over 65% of blade length. Due to this
"over-testing" or "over-stressing" of the inboard blade area, structural damage increasingly occurs towards the end of the test,
which must be inspected and if necessarily, repaired (DNV-GL, 2015). This can result in delays of several weeks in the process.

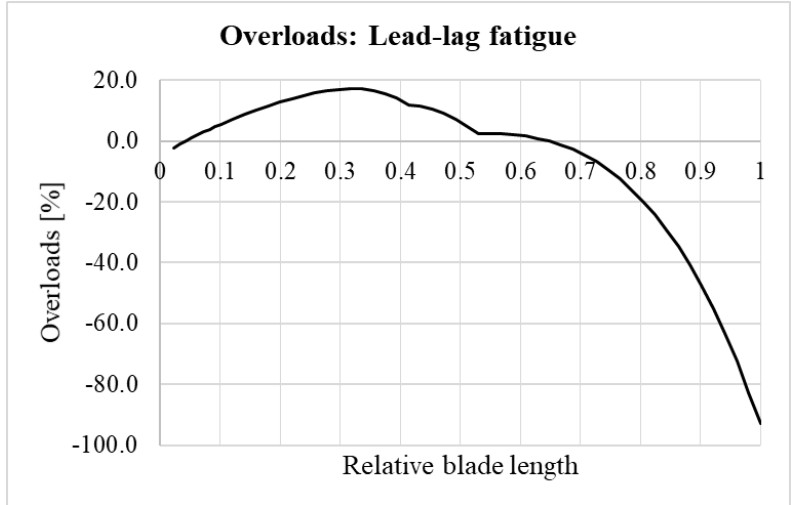


**Figure 3.** Overloads distribution from the full-scale lead-lag fatigue test of 60m blade

Segment test offers a major advantage regarding moment distribution over the blade length. By dividing a full-scale blade into
blade root and blade tip segments, the inboard and outboard area of the rotor blade are decoupled and loaded independently,
as shown in Fig. 4. From the Fig. 4, the variables, M, F, ω, and c represent loadframe mass, force and frequency of cylinder
actuator, and cylinder actuator, respectively. The respective test can be stopped when the corresponding area has reached the
required target loads. Also, should damage occur that necessitates repairs to the blade structure, only the respective segment
test must be stopped for the inspection.

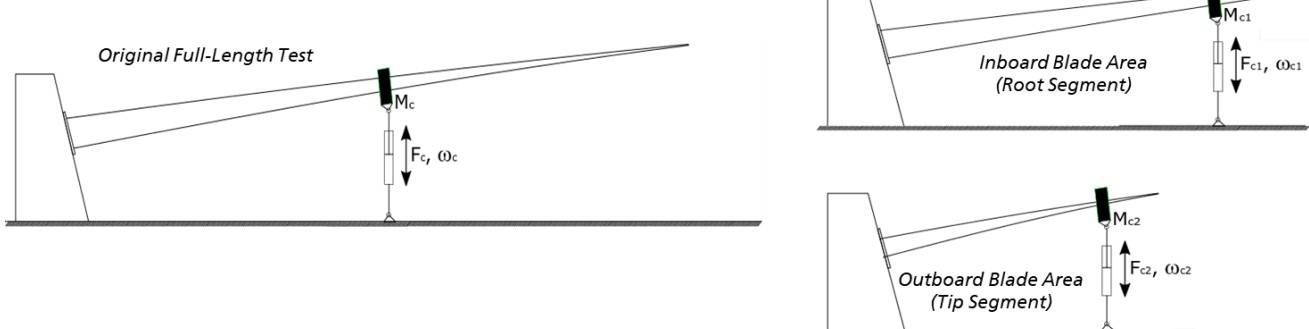

**Figure 4.** Comparison of blade fatigue test setup (full-length blade vs. segment blades)



A widely used method to achieve the introduction of shear forces and bending moment to approximate the bending moment
distribution to the target loads is the attachment of additional masses on the blade. However, this practice leads to a drop in
the test frequency (Spera, 2009; Gasch et. al, 2012). Alternatively, the bending moment distribution can be modified through
artificial stiffness elements, as shown in Fig. 5, by introducing opposing forces and providing additional degrees of freedom
for test quality improvement (Gere et. al, 2013). The stiffness elements can be implemented passively by mechanical or

hydraulic springs or actively by hydraulic actuators (cylinders). In this model, a spring element with a maximum stiffness of
300 kN/m was implemented. In addition, the force entered by the spring into the blade was limited to 100 kN or less. Additional
stiffness elements can also help to increase the systems natural frequency and reduce test time.

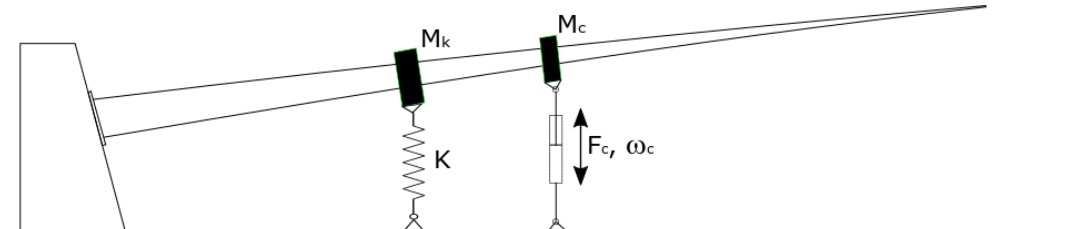

**Figure 5.** Spring element application to blade fatigue test setup

Naturally, the segment test must achieve the same bending moment distribution as the full-length test. However, it is intuitively
obvious that the free end of the segment will experience a zero bending moment. In order to overcome this, a blade dummy,
representing mass and moment of inertia of the tip segment can be fixed to the root section as shown in Fig. 6, which is similar

to the fixation of the blade tip segment to the test bench described above.

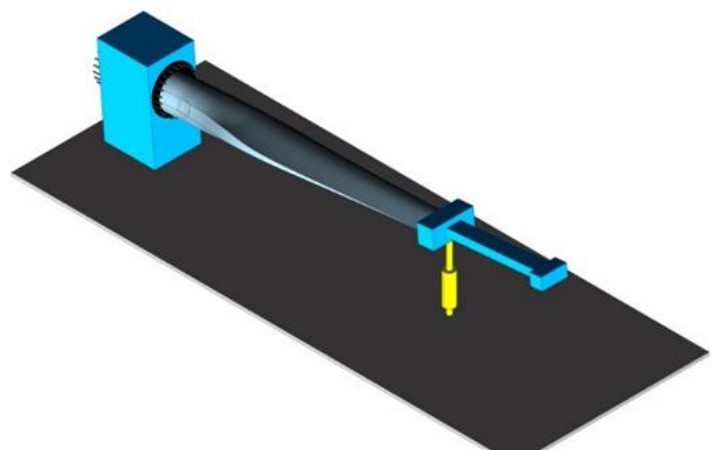

**Figure 6.** Cantilever structure for root segment test





## 2.2 Numerical model for optimization of segment test setup

Purpose of this study is to quantify the advantages of segment blade testing by comparing test setups for two different test cases. For 60m and 90m blade, an optimized test configuration is chosen using dynamic analysis of both blade and test setup.

Rotor diameters of wind turbines installed in 2017 ranged from 48 to 180 meters. The maximum rotor diameter was 142 meters for onshore wind turbine and 180 meters for offshore (Fraunhofer IEE, 2017). In this study, two representative blade models

were selected to represent current onshore and offshore wind turbine rotor blades, shown as blue dots on the scaled trend in Fig. 7. Blades that have been tested at Fraunhofer IWES are also shown in red dots in Fig. 7. Table 1 lists the frequency information of two representative blade models.

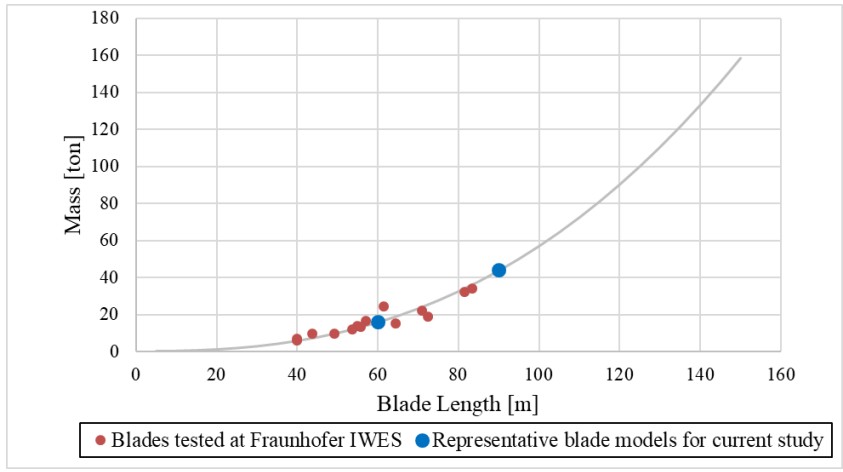

**Figure 7.** Scaled trend of blade mass and length

**Table 1.** Representative blade models

| Blade length | Test frequency in flap | Test frequency in lead-lag |
|:---:|:---:|:---:|
| [m] | [Hz] | [Hz] |
| 60 | 0.71 | 0.86 |
| 90 | 0.49 | 0.82 |

FE beam models of the blades were used for numerical simulation in ANSYS Workbench 18.1 in order to find the optimal test configuration. Fig. 8 shows a schematic representation of the fatigue test model in flap direction and design variables for

optimization analysis. Up to four load frames are included at varying positions along the blade. Each load frame has an associated mass that can be used to tune the frequency and dynamic bending moment distribution. One load frame also acts as the connecting point for a ground based actuator. Frequency and force amplitude of the actuator can be changed as part of the





test configuration. Furthermore, a spring element can be connected to one of the load frames to further modify the system stiffness and natural frequency.

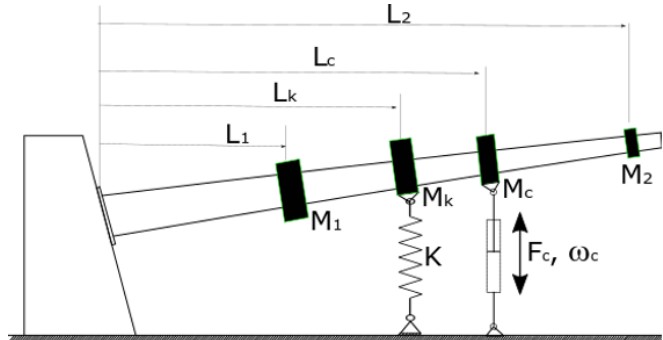

| $L_1, L_2, L_k, L_c$ | Location of load frame (m) |
|---|---|
| $M_1, M_2, M_k, M_c$ | Mass of load frame (kg) |
| K | Spring stiffness (N/m) |
| $F_c$ | Sinusoidal force (N) |
| $\omega_c$ | Actuator frequency (Hz) |


**Figure 8.** Schematic representation of the root segment fatigue test setup

The ranges and bound of the parameter used for numerical simulation in ANSYS are listed in Table 2 and Table 3.

**Table 2.** Range of variable input parameters

| Variable inputs | Lower bound | Upper bound |
|---|---|---|
| $L_1$ | $L_{cr}$ [m] | $(L_{100}+L_{cr})/2$ [m] |
| $L_2$ | $(L_{100}+L_{cr})/2$ [m] | $L_{100}$ [m] |
| $L_k, L_c$ | $L_{cr}$ [m] | $L_{100}$ [m] |
| $M_1, M_2$ | 0 [kg] | 7000 [kg] |
| $M_k, M_c$ | 500 [kg] | 7000 [kg] |
| $\omega_c$ | $0.9 \times \omega_1$ | $1.1 \times \omega_1$ |
| K | 0 [N/m] | $300 \times 10^3$ [N/m] |

∗ $L_{cr}$ : Critical location (location of maximum chord length), ∗ $L_{100}$ : Location of blade tip

∗ $\omega_1$ : 1st eigenfrequency of the test system either in the flap direction or in the lead-lag direction

**Table 3.** Constraints of output parameters

| Contrained parameters | Upper bound |
|---|---|
| $F_c$ | 100 [kN] |
| $F_k$ | 100 [kN] |
| $D_c$ | 1.3 [m] |

∗ $F_c$ : Cylinder actuator force amplitude, ∗ $F_k$ : Spring force, ∗ $D_c$ : Cylinder actuator displacement amplitude






By varying all parameters mentioned above, the system response can be modified to find an optimal test setup. The target function for the optimization process was set to maximize the excitation frequency, while keeping the overload of the bending moment distribution between 0% and 5% for flap and between 0% and 10% for lead-lag test, respectively.

$$\text{Goal}: \text{Max}(\omega_c) \qquad \text{Eq. (1)}$$

$$\text{Constraints}: 0 \leq \text{Overload}\left\{ \frac{(M - M_0^*)}{M_0^*} \right\} \leq 0.05 \ (\text{flap}), 0.1 \ (\text{lead lag}) \qquad \text{Eq. (2)}$$

In Eq. (1) and Eq. (2), M is the bending moment calculated from the segment blade fatigue simulation, and $M_0^*$ is the target bending moment of the full-scale blade. It has to be noted that the constraints are only set for an area of interest, which is defined as ranging from 5% to 50% of the relative blade length for blade root segment, and from division point up to the
relative blade length of 90% in case of blade tip segment.

In order to reduce the computational effort for the optimization, harmonic analysis is used to avoid calculations in the time domain. For the current study, the full harmonic method based on nodal coordinates is used to apply the prescribed displacements support to simulate the realistic test enviroment at a specific test frequency ($\omega_c$)within the range listed in Table
2, and beta damping is used for damping model. As a result, optimization calculations with a large number of individual cases are feasible. Comparison with computations in the time domain have shown that the error introduced by this approach is small. The sequence of this test optimization is shown in Fig. 9. The optimization problem is solved using internal optimization routines provided in ANSYS Workbench. Both gradient based methods and genetic algorithms have proven to give similar results. The optimization parameter used in MOGA (Multi-Objective Genetic Algorithm) is listed in Table 4.

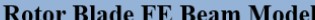

```
┌─────────────────────────────────┐
│     Rotor Blade FE Beam Model   │
└─────────────────────────────────┘
                │
                ▼
┌─────────────────────────────────┐
│      Automated Optimization     │
│  • Generation of different test setups │
│  • Harmonic simulation of fatigue test │
│  • Calculation of respective overload  │
│  • Optimization loop to find best setup│
└─────────────────────────────────┘
                │
                ▼
┌─────────────────────────────────┐
│        Optimal Test Setup       │
│  • Additional masses and their positions │
│  • Excitation: position, frequency, displacement │
│  • Moment and overload distribution │
└─────────────────────────────────┘
```

**Figure 9.** Simulation procedure for test optimization

**Table 4.** Optimization parameters in MOGA

| Optimization parameter | Value |
|---|---|
| Number of initial samples | 1000 |
| Number of samples per iteration | 50 |
| Maximum number of iterations | 20 |


Considering only the pure execution time, test duration can directly be calculated from the actuator frequency and the cycle time. For all considerations in this paper, the number of cycles of the fatigue test are assumed to be 3 million cycles for lead-lag fatigue test and 1 million cycles for flap fatigue test. The test quality is assessed based on the overloading of calculated

bending moments over target bending moments. As addressed above, the overloading is restricted to below 5% for flap fatigue test and 10% for lead-lag fatigue test in the area of interest.





## 3 Feasibility assessment of segment blade fatigue test

Optimized test setups were calculated for the 60m and 90m reference blades using the target functions and optimization constraints above. Test duration and quality as estimated from the numerical simulation are used to evaluate the feasibility of

segment blade fatigue tests.

### 3.1 Test time

The segment blade tests implies parallel testing of root and tip segments. Due to the limitation of root segment length and its lower test frequency than the tip segment, the root segment test is only considered as total fatigue test duration. In a first step, the optimization was conducted without incorporating spring elements. Fig. 10 shows the normalized fatigue test times for

root segment tests at varying segment lengths of the two blades studied. In comparison to the full-length test, the segment test can reduce the test time down up to 67% and 64% for the 60m and 90m blade, respectively. Test time reduction was found to be larger for the 90m blade. Blade separations at 60% of the blade length or shorter did not converge in the optimization, as the criterion for the overload was not matched. This can be explained by the fact that no dummy element as suggested in Fig. 6 was included in the simulations.

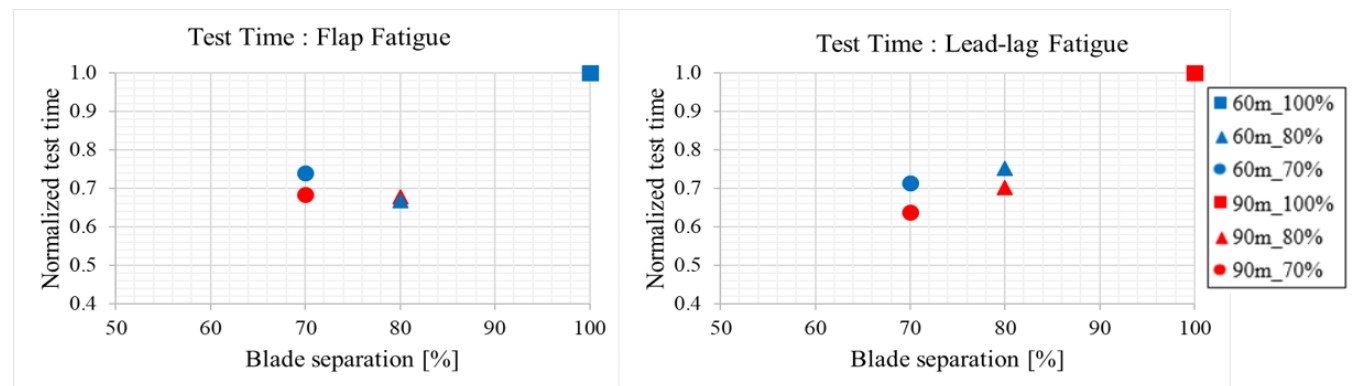


**Figure 10.** Normalized fatigue test time of blade root segments

For the flap fatigue, test time of the 70% root segment is higher than the 80% root segment test time. This is due to the heavy masses added at the outboard of the root segment to meet the target bending moment as explained above.

Figure 11 and Figure 12 show the numerical results from the fatigue test models of 60m and 90m segment blades with spring

elements in comparison to those without spring elements. Depending on the separation point and test direction, the test duration can be reduced by up to 42% more than test cases without stiffness elements. The achievable effects in the lead-lag direction are particularly large.





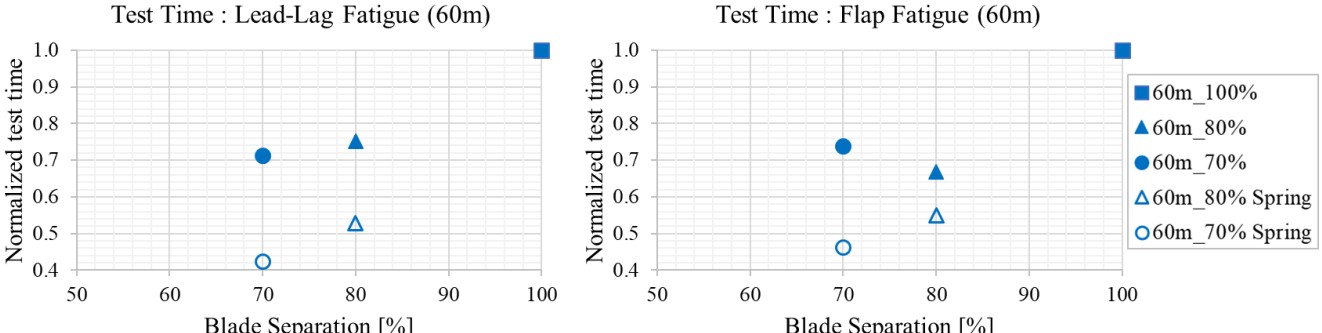

**Figure 11.** Effect of spring elements on fatigue test time of 60m blade root segment

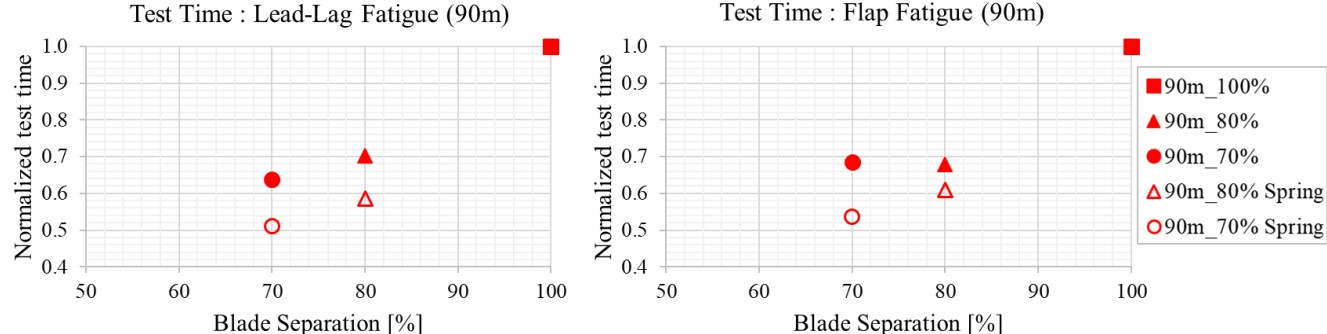


**Figure 12.** Effect of spring elements on fatigue test time of 90m blade root segment

### 3.2 Test quality

Figure 13 shows the overload comparison from the fatigue test in the lead-lag direction of the 60m full-scale blade with segment blades with 80% and 70% separation point, respectively. It shows that segment blade fatigue tests apparently achieve

better test quality compared to the full-scale blade case over the area of interest between 5% to 50% of blade length. In the case of root segment fatigue test, the maximum overloading was less than 10%, while a full-scale case resulted in almost 20% overloads over the observation area.



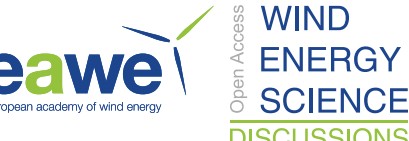

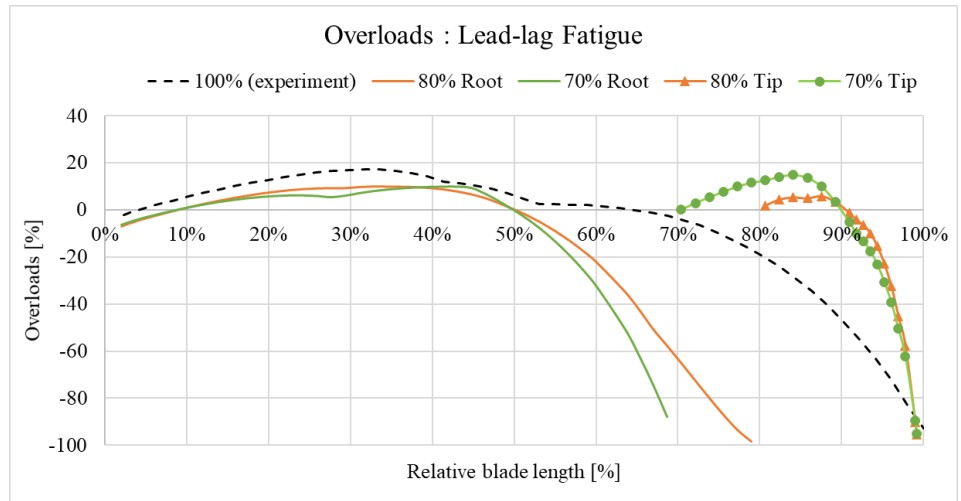

**Figure 13.** Overloads comparison in lead-lag fatigue of 60m blade (full-scale vs. segment)

In the full-length blade test, where all areas are tested simultaneously and at the same frequency, the possibility of sufficiently examining the total blade area is pretty much limited to 65% of the blade length as shown in. The fatigue test of tip segment, however, covers the area from separation point to 90% of blade length sufficiently.

Figure 14 shows the visual comparison of the lead-lag fatigue test setups and under-stressed areas from experimental full-length test with the optimized segment test setup obtained from simulation results. The under-stressed areas are not certifiable

because the target moments were not achieved during fatigue testing. While it is not possible to conclude that the overall area of certifiable regions is significantly increased in the segment blade test, this methodology is able to target regions in a larger range of locations along the blade. Depending on the specific blade design and location of critical regions, this could pose a significant advantage in the certification process.



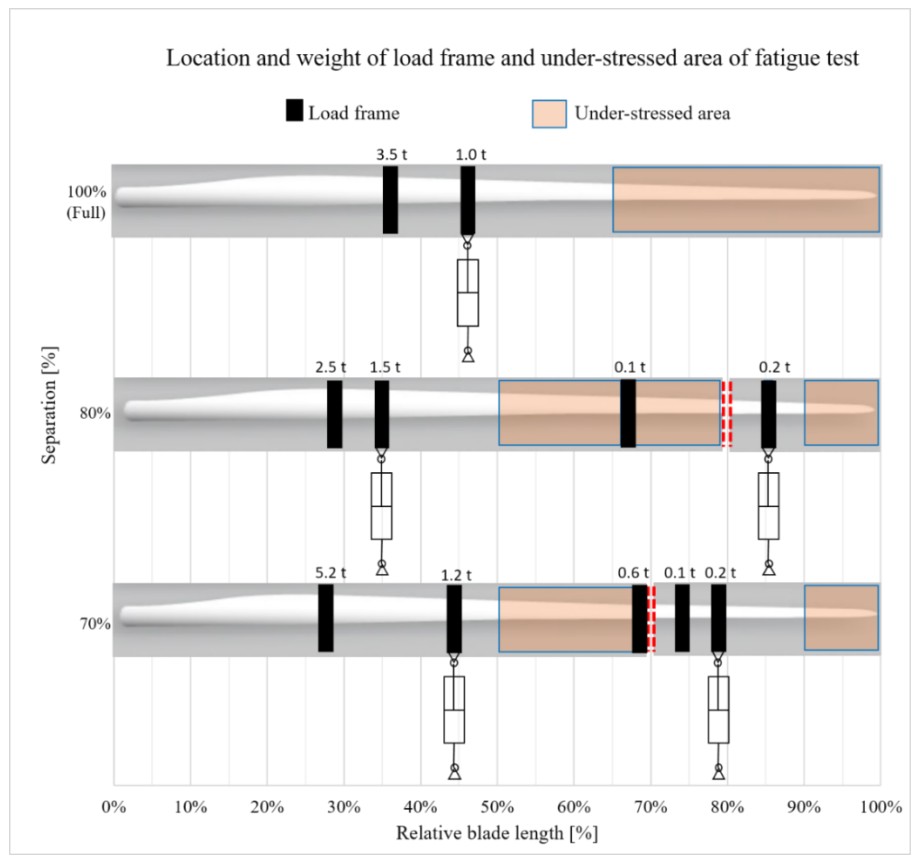

**Figure 14.** Comparison of test setups and under-stressed areas (60m blade, lead-lag fatigue test)

## 4 Conclusions

In this study, a blade division through the separation point at 60% of the blade length or further toward blade root was not feasible without additional structures such as cantilever structure or stiffness elements, since the test quality is poor due to its high overloads. However, he proposed segment test methodology showed the reduction of the total fatigue test time incorporating both flap and lead-lag test up to 72% and 65% for 60m blade and 90m blade, respectively, and further reduction to 43% and 52% with an additional stiffness element in case of 70% segment blade fatigue test compared to full-length fatigue test duration (100%).

The study also showed lower overloads in the areas of interest, compared to the full-length test. Furthermore, it was demonstrated that sufficient loading of the tip section can be achieved. It has to be noted that the optimization goal was to minimize execution time. Using the same numerical models, it is also possible to optimize the setup with respect to minimizing the overload. While this may lead to an increase in test time, it is likely that a larger certifiable region or lower overloads are obtainable.



In conclusion, the study is indicating that segment testing is an effective approach to reduce the duration and increase the test quality of full-scale blade tests – increasingly so for larger turbine blades.


*Author contributions.* KH compiled the literature review, performed numerical simulations, post-processed the data, and wrote the bulk of the paper. MB helped formulate the ideas in the regular discussions. DM perfomend numeric simulations and supported the optimization analysis. SC participated in structuring and review of the paper.

*Data availability.* The data that supports the finding of this research are not publicly available due to confidentiality constraints.

*Competing interests.* The authors declare that they have no conflict of interest.

## 5 Acknowledgements

We acknowledge the support of the within Future Concept Fatigue Strength of Rotor Blades project granted by the German
Federal Ministry for Economic Affairs and Energy (BMWi) (0325939) and the Senator for Health, Environment and Consumer Protection of the Free Hanseatic City of Bremen within ERDF Programme Bremen 2014-2020 (201 / PF_IWES_Zukunfskonzept_Betriebsfestigkeit_ Rotorblatter_Phase I).

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
