# Peer review of "Development and feasibility study of segment blade test methodology"

_Wind Energy Science, 2019_

## Referee Comment (RC1) · Anonymous Referee #1 · 30 Oct 2019

[major questions] 1. In the introduction, the authors said a problem related to the availability of test rigs for testing 100m+ blades due to their large fatigue bending moments, and then introduced a novel segment test methodology. This kind of description may lead to a misconception. The target bending moment at the blade root must be the same, no matter whether a full-length blade is tested or not. For the segment test case, the target bending moment at the root of the root segment is the same as that of the full-length blade. The author should modify the introduction. 2. For the segment testing method, the optimization for the test setups were performed. However, there was no description on the optimization of the overload distribution in Figure 3 for the full-length testing method. Since the authors compared the extents of the overloads for the two testing methods, test setups for the two testing methods must be optimized

first and then the results shall be compared. 3. In Figure 12, for the segment testing method, the lack of the applied fatigue bending moments between the blade length of 50% and 70% is crucial for the blade certification. Since the purpose of the authors' research is developing a new efficient testing method for the blade certification, the authors must suggest a plausible solution about the crucial problem. Without that, it becomes just a paperwork not able to be used in real industry.

[minor comments] Line 15: IEC 61400-23 was published in 2014, not 2012. Line 17: the authors should remove "etc." because in the standard describes the three tests for determining blade properties. Line 18: the official terminology used in the standard is "static tests", not static load test. Line 19: the official terminology used in the standard is "fatigue load tests", not fatigue load test. Line 20: the official terminology used in the standard is "post fatigue static tests", not static load test after fatigue test. Line 84: "the variables, M, F, w, and c", but in Figure 4 the subscript c is not the variable. Line 93: "Error! Reference source not found."

---

## Referee Comment (RC2) · Anonymous Referee #2 · 29 Nov 2019

General comments: very interesting study, which contributes to the scientific progress in wind turbine blade test methodology. Some comments follow, which could improve the understanding of the study and results.

Specific comments: 1. Line 34. Could the author be more specific about which blades have been used for the simulations? 2. Line 96. Motivate why the limit of 300kN/m and 100kN were chosen. 3. Line 186/187. Comment that the test time is lower for 70%, for the lead-lag fatigue results 4. Line 191. Reduction up to 42% referred to the test case without stiffness. It might be better to refer to the full-length test, giving an absolute %. 5. Line 191. Comment the reduction for both 60m and 90m. 6. Figures 11 and 12. Switch the figures, in order to have the same sequence as Fig. 10 (first Flap and second Lead-lag). (Or switch Fig.10, just be coherent). 7. Conclusion. Start with a

short paragraph which explains the purpose of the study.

Technical corrections: 1. In abstract, "time saving up to 43% and 53%", while in conclusion, "time saving up to 43% and 52%". One of them is wrong. 2. Line 42. I would write "The segment testing procedure..." 3. Line 64. "can BE clamped" 4. Line 219. "However, THE...."

———————————————

---

## Author Comment (AC1) · 3 Dec 2019

Dear a Referee,

Thanks for your helpful and qualtitative comments.

Belows are my answers.

Major questions

Comment #1: The author should modify the introduction.

Author feedback #1: I changed the paragraph for clarity as below

Also, testing facilities for 100m+ blades are very limited at this moment. Hence, a

reduction in test time and utilizing available test halls are important to accelerate the development cycle of future offshore turbines with 100m+ blades. In this paper, a novel segment test methodology for wind turbine rotor blades is proposed and its benefits are investigated with a numerical simulation. The proposed test methodology decouples a blade root segment and a tip segment to improve test quality and minimize test time by performing both segment tests independently in parallel while meeting the original target bending moment within specific overload limit (5% for flap direction and 10% for lead-lad direction). ————————————————————- Comment #2: For the segment testing method, the optimization for the test setups were performed. However, there was no description on the optimization of the overload distribution in Figure 3 for the full-length testing method. Since the authors compared the extents of the overloads for the two testing methods, test setups for the two testing methods must be optimized first and then the results shall be compared.

Author feedback #2:I changed plot for comparison between experimental data and optimization result for 100% full-scale blade (60m) with a paragraph belows. (also check the attached plot file). Attached Fig.3 shows it. Also,Fig. 10 shows all optimization results for 100%, 80%, 70%, but not experimental test. Test time between experiment (100%)and optimization result (100%) was already mentioned in Fig. 3. The difference was about 4%. So for better visualization, it was not plotted.

As an example, Fig. 3 shows an experimental overload distribution and an optimized overload distribution of a 60m blade in the lead-lag fatigue test. As a result of experimental fatigue test, the inboard areas of the rotor blade have already experienced the calculated loads from 20 years of life, but the test must continue to sufficiently load the under-stressed areas over 65% of blade length. Due to this "over-testing" or "over-stressing" of the inboard blade area, structural damage increasingly occurs towards the end of the test, which must be inspected and if necessarily, repaired (DNV-GL, 2015). This can result in delays of several weeks in the process. The numerical optimization of a full-length blade fatigue test setup resulted in a better and reduced overload distribution as shown in Fig. 3, but it still shows under-stressed areas after 75% of blade length. Most of all, the increased test time (reverse of test frequency) cannot be ignored as a future offshore blade is getting bigger. Details of an optimization process will be addressed in section 2.2.

Comment #3: In Figure 12, for the segment testingmethod, the lack of the applied fatigue bending moments between the blade length of 50% and 70% is crucial for the blade certification. Since the purpose of the authors' research is developing a new efficient testing method for the blade certification, the authors must suggest a plausible solution about the crucial problem. Without that, it becomes just a paperwork not able to be used in real industry.

Author feedback #3: Some of information related to this are already mentioned in section 2.1, but I updated the paper with some more sentence for your answer as below.

If special interest between 50% and 70% are required, a significantly greater manufacturing outlay like overlapping segments needs to be performed at the separation point of root segment or a blade dummy representing mass and moment inertia of the tip segment could be fixed to the free end of root section as shown in Fig. 6.

———————————— Author feedback on minor questions: Your corrections were very helpful. I corrected everything.

Thanks.

Best regards,

Kwangtae Ha

———————————————————

[Figure]

**Fig. 1.**

---

## Author Comment (AC2) · 3 Dec 2019

Dear a Referee,

Thanks for your helpful and beneficial comments on the paper.

Here are my feedback to your comments.

Comment #1: Line 34. Could the author be more specific about which blades have been used for the simulations?

Author feedback #1: It is not allowed to open more information unfortunately without customer's permission.

_______________________________-

[Figure]

Comment #2: Motivate why the limit of 300kN/m and 100kN were chosen.

Author feedback #2: This is based on applicable range of load frame. Cylinder force can go more than 100kN. But Strucural strength of loadframe was not validated more than 100kN at this moment. So, force (either spring force or cylinder force) was set to 100kN as upper limit for this study based on our experience.
* * *
Comment #3: Line 186/187. Comment that the test time is lower for 70%, for the lead-lag fatigue results

Author comment #3: . For the lead-lag fatigue, test time of the 70% root segment is lower than 80% root segment time because root segment is stiffer over the whole blade length in lead-lag direction than in flap direction, which require less masses at the outboard of the root segment to generate target curvature at the outboard of the root segment.

________________-

Comment #4 & #5 : Line 191. Reduction up to 42% referred to the test case without stiffness. It might be better to refer to the full-length test, giving an absolute %. Line 191. Comment the reduction for both 60m and 90m.

Author feedback #4 & #5: . Depending on the separation point and test direction, the test duration can be reduced up to 43% for a 60m blade and 52% for a 90m blade compared to full-length blades cases without stiffness elements.
* * *
Comment #6 : Figures 11 and 12. Switch the figures, in order to have the same sequence as Fig. 10 (first Flap and second Lead-lag). (Or switch Fig.10, just be coherent).

Author feedback #6: Changed Fig. 10 as suggested (see attached figure)

———————————————

Comment #7: Conclusion. Start with a short paragraph which explains the purpose of the study.

Author feedback #7: This paper proposed a novel segment test methodology for wind turbine rotor blades which mainly aims at improving the efficiency of the fatigue test as a future test method at Fraunhofer IWES.

————————————————————- Technical corrections : I corrected typos and errors as your suggestions.

Thanks for your valuable comments again.

Best regards,

Kwangtae Ha

————————————————————

[Figure]

[Figure]

**Fig. 1.**

---

## Author Response (AR1)

**Development and feasibility study of segment blade test methodology**

Kwangtae Ha[1], Moritz Bätge[1], David Melcher[1] and Steffen Czichon[1]

[1]Fraunhofer Institute for Wind Energy Systems, Department Rotor Blades, Bremerhaven, 27572, Germany

*Correspondence to*: Kwangtae Ha (kwangtae.ha@iwes.fraunhofer.de)

**Abstract.** This paper outlines a novel segment test methodology for wind turbine rotor blades. The segment test mainly aims at improving the efficiency of the fatigue test as a future test method at Fraunhofer IWES. While resulting in reduced testing times, target fatigue bending moments over the intended test area have to be matched within an acceptable range of overload. The numerical simulation reveals that the proposed segment testing has a significant time saving up to 43% and 53% for 60m and 90m blades. When compared to the experimental data of 60m full-length blade fatigue test, the proposed methodology also shows better test quality over the intended area and better certifiable regions over the outboard area.

**1 Introduction**

Wind turbine rotor blades are designed to sustain the designed 20 ~ 25 years lifetime without structural failure. Reliability of the blade is essential for keeping maintenance and operations costs low and maintaining the designed power performance. Mechanical full-scale tests are the main process available for validating the structural integrity of wind turbine rotor blades (Spera, 2009). This is also required as part of the blade certification process according to IEC 61400-23 (IEC, 2014). A typical full-scale blade test consists of following test programs:

> • Determining blade properties such as mass, center of gravity, and natural frequencies.
> • Static tests
> • Fatigue load tests
> • Post fatigue static tests

Among all structural tests, the fatigue load test is the most time-consuming process, and therefore this step accounts for most of the test cost involved. It is executed either in the flap or lead-lag direction independently, or simultaneously in both directions. Due to the large mass and low stiffness of the blades, test operating frequencies are very low, especially for flap fatigue tests. Therefore, fatigue tests take up to several months to accomplish the required cycle count of 1 to 5 million cycles, depending on the blade characteristics (Wingerde et al., 2015; Post, 2016). As blades are getting longer and bigger, both test time and costs increase significantly - especially for future offshore blade measuring more than 100m in length. Also, testing facilities for 100m+ blades are very limited at this moment. Hence, a reduction in test time and utilizing available test halls are important to accelerate the development cycle of future offshore turbines with 100m+ blades.

**메모 포함[하1]:** Response on minor comment #1 (2012 → 2014)

**메모 포함[하2]:** Response on minor comment #2 (Based on IEC 61400-23, 2014, followed the written paragraph in test program in page 16, chapter 7.)

[revised manuscript text omitted]

**메모 포함[하5]:** Response on major question #2-1
Optimization for a full-length blade fatigue test setup showed better overload compared to commercial (experimental) full-blade test. However, test speed was increased instead. Applying the optimization method to segment test concept will be beneficial to save test time while keeping overload to some extent.

**메모 포함[하6]:** Response on minor comment
M,F,w are variables, but not c.

[revised manuscript text omitted]

메모 포함[하8]: Response on major question #2-2
Experimental test time was not plotted in Fig. 10.
Test time between experiment and optimization results was already mentioned in Fig. 3. The difference was about 4%. So for better visualization, it was not plotted.

[revised manuscript text omitted]

메모 포함[하4]: Reply to comment #6
Swich the figures like Fig. 11 and Fig. 12

메모 포함[하5]: Reply to comment #3

메모 포함[하6]: Reply to comment # 4 and # 5
% relative to full-length case both for 60m and 90m

[revised manuscript text omitted]